# Targeting Hypoxia Sensitizes TNBC to Cisplatin and Promotes Inhibition of Both Bulk and Cancer Stem Cells

**DOI:** 10.3390/ijms21165788

**Published:** 2020-08-12

**Authors:** Andrew Sulaiman, Sarah McGarry, Jason Chambers, Emil Al-Kadi, Alexandra Phan, Li Li, Karan Mediratta, Jim Dimitroulakos, Christina Addison, Xuguang Li, Lisheng Wang

**Affiliations:** 1Department of Biochemistry, Microbiology and Immunology, Faculty of Medicine, University of Ottawa, 451 Smyth Road, Ottawa, ON K1H 8M5, Canada; asula097@uottawa.ca (A.S.); smcga078@uottawa.ca (S.M.); jcham018@uottawa.ca (J.C.); ealka095@uottawa.ca (E.A.-K.); aphan055@uottawa.ca (A.P.); ll3@uottawa.ca (L.L.); kmedi072@uottawa.ca (K.M.); Jim.Dimitroulakos@rmp.uhn.ca (J.D.); caddison@ohri.ca (C.A.); sean.li@canada.ca (X.L.); 2Ottawa Institute of Systems Biology, University of Ottawa, 451 Smyth Road, Ottawa, ON K1H 8M5, Canada; 3Department of Basic Science, Kansas City University of Medicine and Bioscience, 1750 Independence Ave, Kansas City, MO 64106, USA; 4Centre for Cancer Therapeutics, Ottawa Hospital Research Institute, Ottawa, ON K1H 8L6, Canada; 5Centre for Biologics Evaluation, Biologics and Genetic Therapies Directorate, Health Canada, Sir Frederick G. Banting Research Centre, 251 Sir Frederick Banting Driveway, Ottawa, ON K1A 0K9, Canada; 6Regenerative Medicine Program, Ottawa Hospital Research Institute, Ottawa, ON K1H 8L6, Canada

**Keywords:** triple negative breast cancer, cancer stem cell, hypoxia, EGFR, cisplatin, PDX

## Abstract

Development of targeted therapies for triple-negative breast cancer (TNBC) is an unmet medical need. Cisplatin has demonstrated its promising potential for the treatment of TNBC in clinical trials; however, cisplatin treatment is associated with hypoxia that, in turn, promotes cancer stem cell (CSC) enrichment and drug resistance. Therapeutic approaches to attenuate this may lead to increased cisplatin efficacy in the clinic for the treatment of TNBC. In this report we analyzed clinical datasets of TNBC and found that TNBC patients possessed higher levels of EGFR and hypoxia gene expression. A similar expression pattern was also observed in cisplatin-resistant ovarian cancer cells. We, thus, developed a new therapeutic approach to inhibit EGFR and hypoxia by combination treatment with metformin and gefitinib that sensitized TNBC cells to cisplatin and led to the inhibition of both CD44+/CD24− and ALDH+ CSCs. We demonstrated a similar inhibition efficacy on organotypic cultures of TNBC patient samples ex vivo. Since these drugs have already been used frequently in the clinic; this study illustrates a novel, clinically translatable therapeutic approach to treat patients with TNBC.

## 1. Introduction

Breast cancer is one of the leading causes of cancer-related deaths in women throughout the world [1]. The triple-negative breast cancer (TNBC) subtype is characterized as negative for the estrogen receptor 1 (ESR1), progesterone receptor (PGR), and human epidermal growth factor receptor type 2 (HER2).

In contrast to other breast cancer subtypes, TNBC does not have specific targets and is reliant on chemotherapeutic regiments for systemic treatment and disproportionately accounts for the majority of breast cancer related deaths. Cisplatin is a platinum 2 complex capable of creating adducts, causing DNA damage, and subsequently inducing apoptosis in a multitude of cancers [2]. Recent clinical trials have showed efficacy of cisplatin in combinational chemotherapy in comparison to conventional chemotherapeutic approaches for the treatment of TNBC [3,4,5]. However, others have demonstrated lackluster outcomes, and underlying mechanisms remain convoluted [6,7].

Hypoxia following cisplatin treatment has been reported and associated with the onset of drug resistance and disease progression [8]. Additionally, while chemotherapy effectively inhibits bulk tumor populations, it enriches cancer stem cells (CSCs) [9]. CSCs retain stem cell-like characteristics and are highly associated with drug resistance, metastasis and tumorigenicity [10,11]. In breast cancer, two main populations of CSCs have been characterized according to the expression of CD44+/CD24− (mesenchymal-like CSC) or ALDH+ (epithelial-like CSC) [10,12,13]. Two populations of CSC exhibited plasticity and strong tumorigenic capacity, capable of generating new tumors and responsible for disease relapse. Thus, a new strategy to simultaneously target both CSC populations and suppress hypoxia will lead to the effective treatment of TNBC.

EGFR signaling has been shown to enhance aerobic glycolysis in TNBC cells to promote tumor growth, which has been considered as a potential target for TNBC treatment [14,15]. Gefitinib, an EGFR inhibitor, elicits anti-hypoxic properties and has been demonstrated to overcome cisplatin-induced hypoxia resistance in TNBC [16]. However, gefitinib is prone to resistance via stimulation of bypass signalling which in turn promotes Akt/mTORC1 and subsequent hypoxia, diminishing the efficacy of gefitinib for long-term treatment [17,18,19]. To circumvent this, metformin (an AMPK activator and first line treatment for type II diabetes) has been employed. Metformin exhibits anti-hypoxic effects via inhibition of mTORC1 activity and HIF-1α protein stability [20,21]. Metformin has been shown to reverse gefitinib hypoxia resistance in head and neck squamous cell carcinoma and attenuate hypoxia-induced cisplatin resistance; moreover, metformin has been found to target CSCs in TNBC [22,23,24]. However, whether the combination of metformin and gefitinib can prevent cisplatin-induced hypoxia in TNBC, increase apoptosis and inhibit CSCs has not yet been reported.

In this report, we found that TNBC tumors from patients expressed high levels of EGFR and hypoxia genes compared to normal mammary tissue in a clinical database. As such, we developed a combinational therapy consisting of metformin (AMPK activator), gefitinib (EGFR inhibitor) and cisplatin. We found that inhibition of EGFR and hypoxia sensitized TNBC cell lines to cisplatin treatment. Both ALDH+ epithelial and CD44+/CD24− mesenchymal CSC populations could be inhibited. These findings were further verified using ex vivo organotypic cultures of TNBC clinical samples and TNBC patient-derived xenograft tumors. Together, these findings present a new strategy to prevent cisplatin-induced CSCs and hypoxia, which may lead to an efficacy treatment to reduce disease relapse in TNBC patients.

## 2. Results

### 2.1. Upregulated Gene Expressions of EGFR and Hypoxia Associate with Cisplatin Resistance, Anti-Apoptosis, and Stemness

Clinical trials showed mixed results for the treatment of TNBC with cisplatin with convoluted mechanisms [3,4,5,6,7]. To identify potential targets associated with the cisplatin resistance (cisplatin sensitive and resistant) due to unavailable datasets for TNBC, we adopted an ovarian cancer bioinformatics model. We assessed gene overexpression in cisplatin resistant vs. naive ovarian cancer cells using the NCBI Gene Expression Omnibus (GEO2R). It was found that cisplatin resistance was associated with the increased expressions of EGFR, hypoxia, stemness and anti-apoptotic genes (Appendix A). We then analyzed NCBI Gene Expression Omnibus (GEO2R) to compare 30 TNBC with 13 normal mammary tissue samples and identified a similar trend where TNBC samples possessed high levels of EGFR, hypoxia, stemness and anti-apoptotic gene expressions compared to normal mammary tissues (Figure 1A–D). EGFR and hypoxia have both been associated with TNBC resistance to cisplatin [25,26], suggesting potentially clinical targets for further investigation.

### 2.2. Combination of Hypoxia and EGFR Inhibitors with Cisplatin Effectively Suppress TNBC Bulk and CSC Populations

To determine whether hypoxia and EGFR inhibition could increase cisplatin efficacy, 25 µM of metformin (an AMPK activator used to treat type II diabetes with anti-hypoxic effects), 5 µM gefitinib (an EGFR inhibitor), and 5 µM of cisplatin at clinically relevant concentrations were used in different combinations (designated as CMG). As shown in Figure 2A,B, CMG treatment for 120 h significantly reduced bulk cell viabilities in both SUM 149-PT and MDA-MB-231 TNBC cell lines.

To assess whether the CMG combinational therapy could also target both epithelial-like and mesenchymal-like CSC populations, we preformed FACs analysis on SUM 149-PT and MDA-MB-231 cells after 120 h of treatment. CSCs have been closely associated with drug resistance, cancer metastasis, and poor prognosis and chemotherapeutics enriched CSCs [9]. There are two main CSC populations characterized in TNBC: CD44+/CD24− mesenchymal and ALDH+ epithelial CSCs. Two CSC subpopulations exhibit plasticity and 100-fold greater tumor-initiating capacity than their non-CSC counterparts [11,27].

We found that CMG effectively inhibited CD44+/CD24− CSC population in both SUM 149-PT and MDA-MB-231 TNBC cell lines and reduced the living CSCs by ~90% (Figure 2C,D,G, Appendix A). In contrast, single and dual treatments exhibited weak or mixed results. Additionally, gefitinib alone did not inhibit CD44+/CD24− CSCs in MDA-MB-231 cells although it showed efficacy in SUM149-PT cells.

For the ALDH+ CSC subpopulation, we found that the combinations of cisplatin+gefitinib, metformin+gefitinib and CMG were all effective (Figure 2E,F,H, Appendix A) with CMG reducing ALDH+ CSCs in both TNBC cell lines by ~90%. Together, these data demonstrate that the CMG combination can effectively inhibit TNBC bulk, CD44+/CD24− mesenchymal CSCs and ALDH+ epithelial CSC populations in both MDA-MB-231 and SUM149-PT cell lines at clinically relevant concentrations.

### 2.3. Combination of Metformin, Gefitinib and Cisplatin Effectively Repress Hypoxia and Promote Apoptosis in TNBC

As hypoxia has been closely associated with CSC enrichment and chemoresistance in breast cancer, we asked whether combinational CMG treatment suppressed hypoxia [28,29]. Hypoxia was assessed using a luciferase reporter plasmid containing three hypoxia response elements from the Pgk-1 gene upstream of firefly luciferase [30]. Using this system, active HIFs would bind to hypoxia response elements to mediate transcription and induce luciferase transcription [31]. We found that cisplatin increased the transcriptional activity of HIFs while the combination of metformin and gefitinib (MG) reduced HIF activity (Figure 3A). Moreover, the combination of CMG was able to antagonize cisplatin-upregulated HIF activity (Figure 3A). Similar findings were observed using a HIF1α oxygen-dependent degradation sequence luciferase reporter (ODD-Luc) hypoxia reporter (Figure 3B) [32]. Additionally, RT-qPCR analysis of hypoxia-response-element containing genes showed that CMG treatment suppressed PDK1 and LDH1 gene expression (Figure 3C,D) [33,34].

FACs analysis was preformed to assess whether inhibition of hypoxia with CMG would correlate with a sensitization of cells to cisplatin-mediated apoptosis. Individual and dual combinations of cisplatin, metformin and gefitinib did not significantly induce early apoptosis (Annexin-V+/7-AAD−) or late apoptosis (Annexin-V+/7-AAD+) in TNBC cells (Figure 3E). Early apoptotic cells represent the cells undergoing apoptosis while retaining an intact cell membrane at the time of analysis. Notably, only the CMG combination significantly increased both early and late apoptosis in MDA-MB-231 cells (Figure 3E). Together, these data suggest that the CMG combinational treatment is able to inhibit hypoxia and promote TNBC apoptosis.

### 2.4. CMG Treatment Effectively Inhibits the Viability of TNBC Organotypic Cultures and Reduces Their CSC Subpopulations

To verify the above findings in a clinically relevant model, we obtained two clinical samples (CRDCA and SEM-1) from patients with TNBC and three TNBC patient-derived xenograft (PDX) tumors (HCI-001, HCI-002 and HCI-016). Samples were sliced into 2 × 1 mm, cultured ex vivo and treated with the drug in different combinations at the clinically relevant concentrations for 144 h. Alamar blue viability analysis demonstrated the efficacy of CMG treatment in all five clinically heterologous samples, showing a significant reduction in cell viability post-treatment (Figure 4A). Moreover, CMG combination also significantly suppressed both epithelial (ALDH+) and mesenchymal (CD44+/CD24−) CSC subpopulations in all five clinically heterologous samples (Figure 4B–E).

Of note, the results from clinical TNBC samples (three PDX and two clinical samples) amongst the single drug inhibitors are not same as the results from TNBC cell lines (as shown in Figure 3). Indeed, the gene expressions of EGFR, apoptotic resistance, stemness and hypoxia between TNBC patient tumors and TNBC cell line were different in the NCBI Gene Expression Omnibus (GEO2R) (Appendix A). These differences also highlight the importance to use clinically translatable models such as PDX for the development of therapeutic interventions.

## 3. Discussion

Cisplatin is a chemotherapeutic drug and has been widely used for the effective treatment of ovarian, cervical, bone, esophageal and lung cancers [35,36,37,38,39,40]. Renewed interest in cisplatin has led to more clinical trials, showing efficacious results in the treatment of other types of cancer, including TNBC [41,42,43,44,45]. Clinical trials using cisplatin for the treatment of TNBC discourage the use of cisplatin alone as adjuvant or neoadjuvent therapy [7]. However, cisplatin in combination with gemcitabine/docetaxel or bevacizumab in clinical trials has shown better outcomes [3,46,47]. Trial CBCSG006 found that cisplatin and gemcitabine combination displayed superior efficacy in comparison to the conventional paclitaxel and gemcitabine combination for the treatment of metastatic TNBC [41].

A drawback for cisplatin treatment is its modulation of an assortment of signalling pathways that bars its long-term efficacy. Cisplatin has been found to induce interleukin-6 (IL-6) which promotes the accumulation of hypoxia inducible factors and CSC enrichment [8,48]. Cisplatin has also been demonstrated to induce STAT3 and NF-kβ expression following treatment. Of note, IL-6 triggers STAT3 and NF-kB signaling pathways in TNBC [49,50]; both STAT3 and NF-kB pathways are direct regulators to stimulate HIF-1α, chemoresistance and CSC enrichment [51,52,53]. With the expanding role of cisplatin in TNBC treatment, suppressing cisplatin-induced hypoxia is critical for preventing cisplatin-resistance and CSC enrichment following treatment [8,54,55,56].

To prevent cisplatin-induced hypoxia, we employed metformin (first-line therapy for the treatment of type II diabetes), which was found to potently suppress hypoxia [21,57,58,59]. Metformin inhibits complex I of the mitochondria to indirectly stimulate AMPK, which, in turn, activates TSC1/TSC2 to inhibit mTORC1 and subsequent hypoxia signalling. Mechanistic analysis of metformin in an esophageal cancer cell model demonstrated a widespread downregulation of P13K/mTOR related proteins and stemness-related proteins, which correlated with a reduction in ALDH+ CSCs and increased apoptosis when combined with chemotherapy [60]. However, a study using AMPKα siRNA on HEP2G and Huh7 liver cancer cell lines showed that AMPK activation did not affect HIF-1α. Interestingly, treatment with metformin was still able to potently suppress HIF-1α protein levels and transcriptional activity via AMPK independent modulation of HIF1α protein stability [21]. Thus, metformin exhibits anti-hypoxic properties through AMPK dependent and independent mechanisms.

Short-term Gefitinib treatment prevents EGFR activity and subsequently activate Akt/mTORC1, leading to inhibition of cisplatin-induced hypoxia [16,61]. Long-term gefitinib treatment promotes Akt/mTORC1 signalling via activation of STAT3, PLAUR and/or TGF-α, which leads to hypoxia, CSC related gene expression and gefitinib resistance [18,62]. We found that even short-term treatment of TNBC cells with gefitinib led to the upregulation of hypoxia-related PDK1 gene, a potential early sign of resistance (Figure 3C).

Since gene expressions of hypoxia and EGFR were upregulated in TNBC patient tumors in comparison to normal mammary tissue (Figure 1A,B), we reasoned that their inhibition may increase cisplatin efficacy in TNBC. In addition, upregulation of hypoxia and EGFR is also associated with cisplatin resistance in ovarian cancer models (Appendix A), which further supports this possibility. Indeed, CMG combination effectively reducing TNBC cell viability and inhibiting both mesenchymal CD44+/CD24− CSCs and epithelial ALDH+ CSCs (Figure 2A–H and Appendix A). Luciferase-gene reporter analysis demonstrated that cisplatin promoted TNBC hypoxia (Figure 3A–C), which may associate with drug resistance and CSC enrichment. However, CMG combination was able prevent cisplatin-induced hypoxia and promote apoptosis (Figure 3A–E).

Using organotypic cultures of five TNBC patient tumor fragments, we further verified our findings. Importantly, we revealed a differential pattern of gene expressions between TNBC patient tumors TNBC cell lines (Appendix A). This highlights the importance to use ex vivo patient tumors in experiments or in addition, in vivo patient tumors in mouse models, for clinical translation. Of note, the patient tumors from PDX samples were selected to represent a clinical setting, including paclitaxel-resistant HCI-001, paclitaxel-sensitive HCI-002 and naïve HCI-016 (obtained from patient without prior exposure to chemotherapy) [63,64]. Importantly, combination of CMG at clinically relevant dosages was capable of inhibiting both bulk and CSC subpopulations in all five clinical tumors in ex vivo organotypic cultures (Figure 4A–E) [65,66,67]. As assessment of drug efficacy using short-term organotypic cultures of PDX have demonstrated clinical concordance [68,69,70,71], our data support the clinical translatability of the CMG treatment. As cisplatin, gefitinib, and metformin are all currently used in clinic, our study may lead to a new avenue in TNBC treatment.

## 4. Materials and Methods

### 4.1. Cell Culture and Reagents

MDA-MB-231 breast cancer cells were purchased from the American Type Culture Collection (ATCC, Manassas, VA, USA) and maintained in DMEM-F12 media supplemented with 10% Fetal bovine serum (FBS, HyClone, Logan, UT, USA) and 1% penicillin/streptomycin. SUM 149-PT breast cancer cells were obtained from Asterand (Detroit, MI, USA) and cultured in Hams F-12 media (Mediatech, Manassas, VA, USA) containing 5% FBS, 5 μg/mL insulin, 1 μg/mL hydrocortisone, 10 mm HEPES, and 1% penicillin/streptomycin. Cells were cultured at 37 °C in a 5% CO2 incubator. Cisplatin was purchased from Caymen Chemical Company (Ann Arbor, MI, USA), gefitinib from LC Labs- G-4408 (Woburn, MA, USA) and metformin from Caymen Chemical Company (Ann Arbor, MI, USA).

### 4.2. Organotypic Cultures of Patient TNBC Breast Tumor and Patient-Derived Xenograft Fragments

Tumor tissues from two TNBC patients undergoing routine surgical procedures were obtained following the protocol approved by The Ottawa Hospital Research Ethics Board (Protocol# 20120559-01H). Approximately 2 mm cores were obtained using a sterile biopsy punch that was further sliced with a scalpel to obtain approximately 2 × 1 mm tumor slices. The slices were randomized and placed into a well of 24-well plate and cultured in DMEM-F12 medium supplemented with 10% FBS, 1% penicillin/streptomycin, 1 µg/mL insulin, 0.5 ng/mL hydrocortisol and 3 ng/mL epidermal growth factor. The tissue fragments were treated with the same concentrations of inhibitors as described in the figures, followed by a viability assay and flow cytometric analysis as previously described [72,73]. The TNBC patient-derived xenograft samples HCI-001, HCI-002, HCI-016 was obtained from University of Utah, cut similarly to the patient tumor samples and cultured in the same conditions as the clinical samples.

### 4.3. Flow Cytometry Analysis

Dissociated cancer cells were filtered through a 4 µm strainer and suspended in PBS supplemented with 2% FBS and 2 mM EDTA (FACS buffer) as previously described [73]. One µL of mouse IgG (1 mg/mL) was added and incubated at 4 °C for 10 min. The cells were then re-suspended in 1× binding buffer and anti-CD44 (APC) in combination with anti-CD24 (PE) antibodies (BD, Mississauga, ON, Canada) according to the manufacturer’s instructions for 30 min. The cells were washed twice with FACS buffer and 7-aminoactinomycin D (7-AAD, eBioscience, San Diego, CA, USA) and Annexin-V/V450 (BD) was added and incubated for 15 min at room temperature to assess dead and apoptotic cells. Flow cytometry was performed on the BD LSRFortessa. Data were analyzed with FlowJo software (Ashland, OR, USA).

### 4.4. Quantitative Real-Time PCR

Total RNAs were extracted using RNeasy kit (QIAGEN) and real-time qPCR (RT-qPCR) analysis was performed using Bio-Rad MyiQ (Bio-Rad, Hercules, CA, USA) as described previously [72]. The conditions for RT-qPCR reactions were one cycle at 95 °C for 20 s followed by 45 cycles at 95 °C for three seconds and annealing at 60 °C for 30 s. Results were normalized to the housekeeping gene 18S ribosomal RNA (18S). Relative expression level of genes from different groups were calculated with the 2^−ΔΔ*C*t^ method and compared with the expression levels of appropriate control cells. Specific primer sequences for individual genes are listed in Table 1.

### 4.5. Luciferase Reporter

MDA-MB-231 TNBC cells were seeded into 24-well plates and transfected with 500 ng of ODD-Luciferase-pcDNA3 (Addgene Plasmid # 18965, a gift from William Kaelin) [32], or HRE-luciferase (Addgene Plasmid # 26731, a gift from Navdeep Chandel) [73] in conjunction with 500 ng Renilla pRL-SV40P construct (Addgene Plasmid #27163, a gift from Dr. Ron Prywes) [74] using Lipofectamine 2000 (Invitrogen) according to the manufacturer’s instructions. After 18 h, cells were treated with either DMSO (vehicle), cisplatin (5 µM), metformin (25 µM), gefitinib (5 µM) or different combinations for 24 h, after which cells were lysed and both Firefly and Renilla luciferase activities were quantified using a Dual-Luciferase^®^ Reporter Assay System (Promega, Madison, WI, USA) following the manufacturer’s instructions.

### 4.6. Cell Viability Assays

MDA-MB-231 cells were seeded into 24 well plates (5 × 104 cells/well). After 120 h of treatment, 3-(4,5-dimethylthiazol-2-yl)-2,5-diphenyl tetrazolium bromide (MTT, 5 mg/mL) was added to determine cellular viability. Absorbance was read at 570 nm. For clinical/PDX tumors, Alamar blue viability analysis was performed by incubation with 10% Alamar blue reagent (Thermo Fisher Scientific, Waltham, MA, USA) for 4 h. Florescence was measured at 560 nm excitation and 590 nm emission. Alternatively, cell counting was preformed after trypsinization of cells into single cell suspension followed by 1:10 dilution with trypan blue dye, followed by counting viable cells in haemocytometer under microscopy.

### 4.7. Clinical Database Analysis

Gene Expression Omnibus2R database [75,76] was used to analyze various datasets. Dataset: GSE15709 was used to compare cisplatin resistant and sensitive ovarian cancer cell lines [77]: https://www.ncbi.nlm.nih.gov/geo/query/acc.cgi?acc=GSE15709 (Appendix A). Dataset GSE38959 was used to compare 30 TNBC patient samples to 13 normal mammary tissue samples [78]: https://www.ncbi.nlm.nih.gov/geo/query/acc.cgi?acc=GSE38959. Dataset GSE65216 was used to compare 55 TNBC patient samples to 12 TNBC cell lines [79,80,81]: https://www.ncbi.nlm.nih.gov/geo/geo2r/?acc=GSE65216&platform=GPL570.

### 4.8. Statistical Analysis

For all clinical databases, the log rank test was performed to determine the statistical differences between groups. Data are expressed as means +/− standard deviation (SD) or standard error (SE). Data distributions were tested by one-way ANOVA and statistical differences between groups were assessed by unpaired Student’s t-test (comparison of two groups). Results were considered significant when * *p* < 0.05, ** *p* < 0.01, or *** *p* < 0.001.

## Figures and Tables

**Figure 1 ijms-21-05788-f001:**
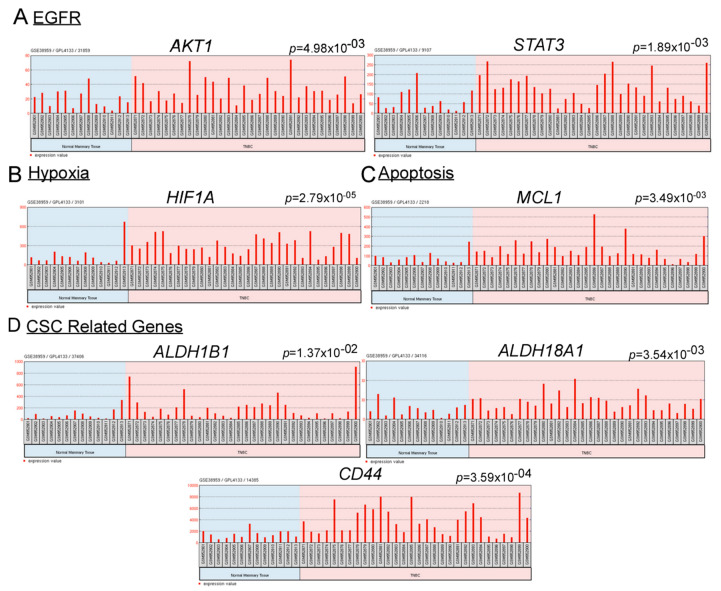
TNBC is associated with high levels of hypoxia, EGFR, stemness-related and anti-apoptosis related gene expression. (**A**–**D**) The relative expression levels (A.U arbitrary unit) of genes in 30 TNBC and 13 normal mammary tissue samples using the NCBI Gene Expression Omnibus (GEO2R). GSE38959 samples were analyzed using the GPL4133 Agilent-014850 Whole Human Genome Microarray 4 × 44K.

**Figure 2 ijms-21-05788-f002:**
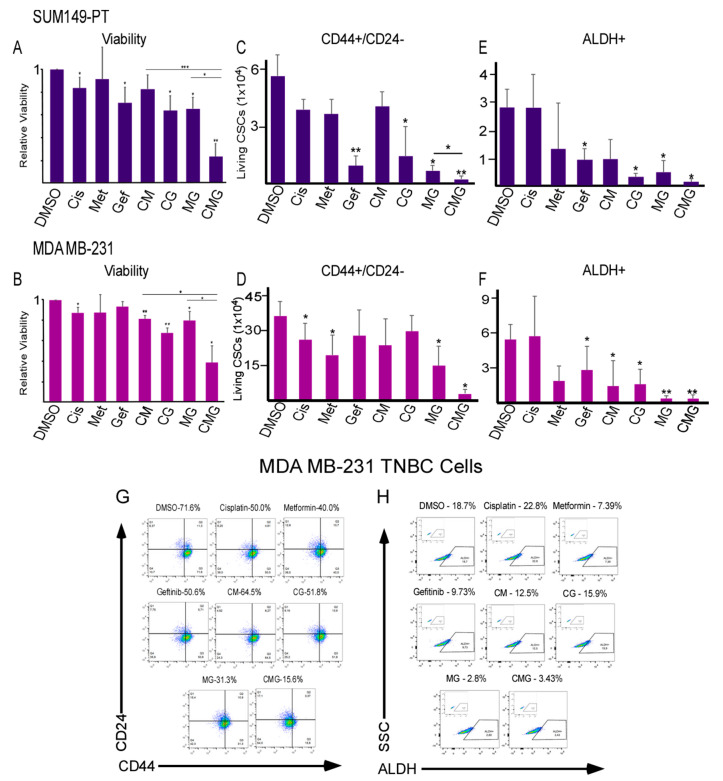
Co-inhibition of hypoxia and EGFR in combination with cisplatin effectively suppress bulk tumor cells and CSC subpopulations in TNBC. (**A**) Relative viability (MTT assay) of SUM 149-PT TNBC cells after 120 h of exposure to DMSO (vehicle control), cisplatin (Cis, 5 µM), gefitinib (Gef, 5 µM) and/or metformin (Met, 25 µM). (**B**) Relative viability (viable cell counting) of MDA-MB-231 TNBC cells after 120 h of the same treatment as described in (A) (**C**,**D**) Living CSCs (CD44+/CD24−) in SUM 149-PT and MDA-MB-231 cells after 120 h of exposure to the drugs as described in (A). (**E**,**F**) Living CSCs (ALDH+) in SUM 149-PT and MDA-MB-231 cells after 120 h of exposure to the drugs as described in (A). (**G**,**H**) Representative flow cytometric plots of the CSC subpopulations (CD44+/CD24− or ALDH+) in MDA-MB-231 cells after 120 h of treatment with the drugs as described in (A). Data represent means ± SD, *n* = 3 repeats; * *p* < 0.05, ** *p* < 0.01, *** *p* < 0.001.

**Figure 3 ijms-21-05788-f003:**
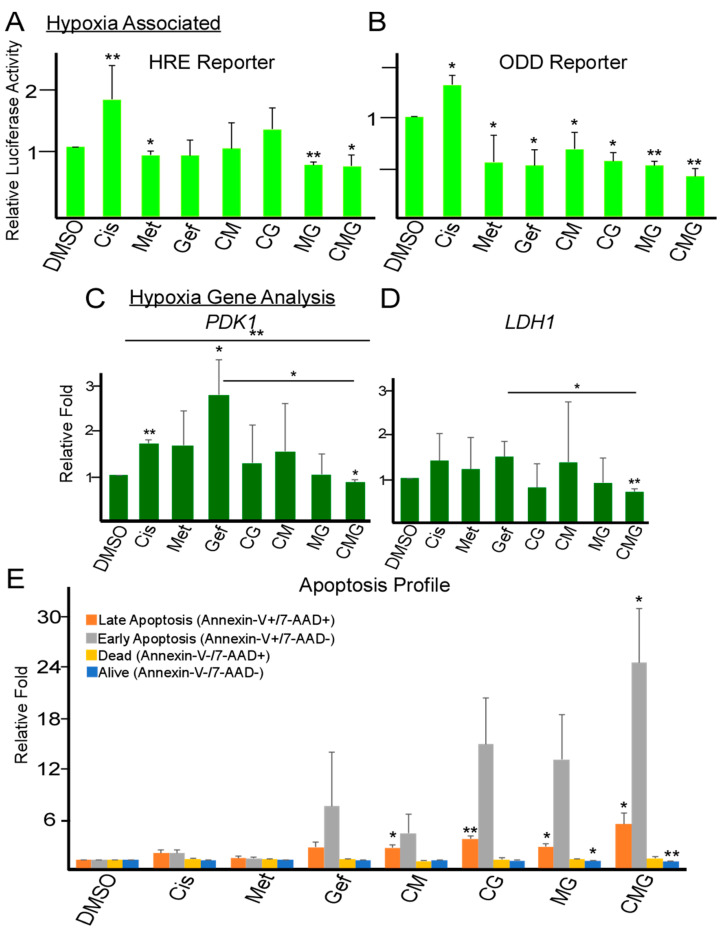
Co-inhibition of hypoxia and EGFR suppresses cisplatin-upregulated hypoxia and promotes apoptosis in TNBC. (**A**,**B**) Relative luciferase reporter expression of MDA-MB-231 TNBC cells after 24 h of drug treatment. Cells were transfected with HRE-Luciferase or ODD-Luciferase and normalized with the pRL-SV40P Renilla construct. The following day, cells were treated with DMSO (vehicle control), cisplatin (Cis, 5 µM), gefitinib (Gef, 5 µM) and/or metformin (Met, 25 µM). Data represent means ± SD, *n* = 3 repeats; * *p* < 0.05, ** *p* < 0.01. (**C**,**D**) RT-qPCR analysis and comparison of relative mRNA expression levels of hypoxia genes (PDK1 and LDH1) after treatment for 48 h as described in (**A**,**B**). Data represent means ± SD, *n* = 3 repeats; * *p* < 0.05, ** *p* < 0.01. (**E**) Flow cytometric analysis of cell states (Dead, Alive, Early Apoptosis or Late Apoptosis) based on Annexin-V/7-AAD markers after 72 h of exposure to DMSO (vehicle control), cisplatin (Cis, 5 µM), gefitinib (Gef, 5 µM) and/or metformin (Met, 25 µM). Data represent means ± SE, *n* = 3 repeats; * *p* < 0.05, ** *p* < 0.01.

**Figure 4 ijms-21-05788-f004:**
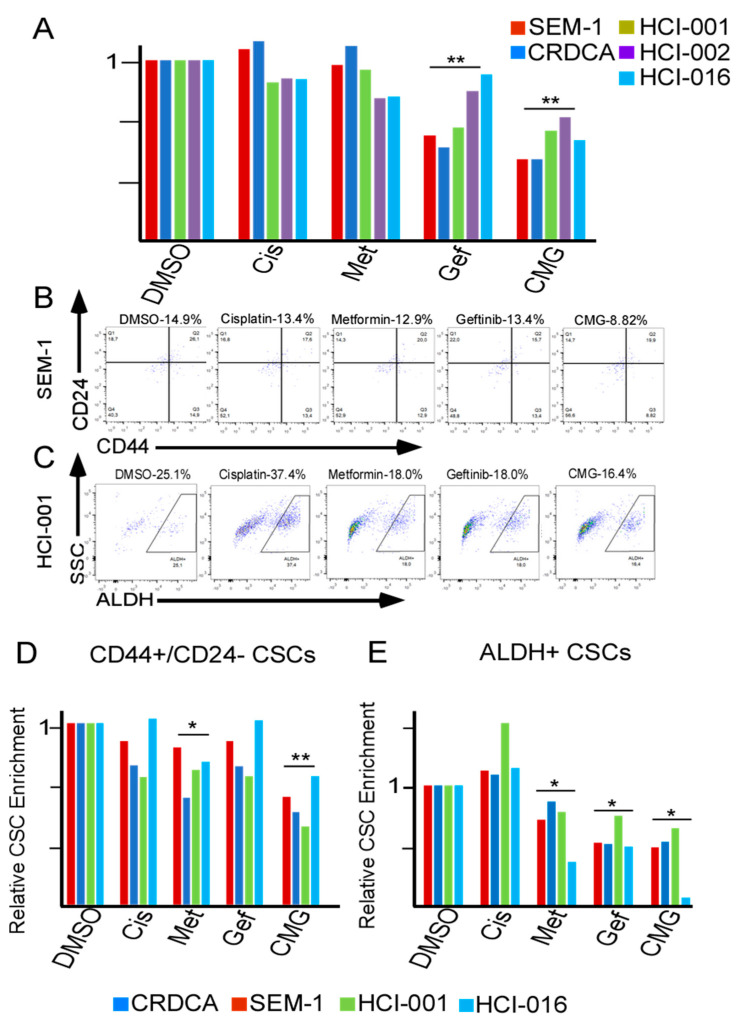
Co-inhibition of hypoxia and EGFR in combination with cisplatin effectively inhibits the viability of patient tumors and CSC populations in ex vivo organotypic cultures. (**A**) Alamar blue viability analysis of organotypic cultures from two primary patient TNBC tumors (CRDCA and SEM-1 samples) and three patient-derived xenograft samples (HCI-001, HCI-002, HCI-016) after 144 h of exposure to DMSO (vehicle control), cisplatin (Cis, 5 µM), gefitinib (Gef, 5 µM) and/or metformin (Met, 25 µM). (**B**,**C**) Representative flow cytometric plots of the CSC subpopulations (CD44+/CD24− or ALDH+) in SEM-1 and HCI-001 cells after 144 h of treatment with the different drugs as described in (**A**). (**D**,**E**) Flow cytometric data of the relative living CSC (CD44+/CD24− or ALDH+) populations in SEM-1, CRDCA, HCI-016 and HCI-001 patient TNBC tumors after 144 h of exposure to the drugs as described in (**A**). Each bar represents individual patient sample, *n* = 4–5 samples; * *p* < 0.05, ** *p* < 0.01.

**Table 1 ijms-21-05788-t001:** Primers used in qPCR.

Genes	Forward	Reverse
18S	AACCCGTTGAACCCCATT	CCATCCAATCGGTAGTAGCG
GAPDH	ACAGTCAGCCGCATCTTCTT	GACAAGCTTCCCGTTCTCAG
PDK1	CAACAGAGGTGTTTACCCCC	ATTTTCCTCAAAGGAACGCC
LDH1	GGCCTGTGCCATCAGTATCT	GGAGATCCATCATCTCTCCC

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
