# Peer review of "Targeting Hypoxia Sensitizes TNBC to Cisplatin and Promotes Inhibition of Both Bulk and Cancer Stem Cells"

_ijms, 2020, doi:10.3390/ijms21165788_

Round 1
Reviewer 1 Report
Review for Manuscript ijms-890003-peer-review-v1
General Comments: Very nicely written and presented. I appreciate the use and extrapolation from the cisplatin-resistant ovarian cancers. Mostly minor comments below and one general comment for Discussion:
1) How does high levels of EGFR, hypoxia, stemness, and anti-apoptotic gene expressions relate to hormone-receptor positive BrCas?
More specific comments are listed below in order since there are no line numbers.
More Specific Comments:
Title – None
Abstract
1) Change “clinical dataset of TNBC” to “clinical datasets of TNBC”
2) Change “combination of” to “combination treatment with”
Introduction
1) When beginning to discuss Gefitinib, introduce the importance of EGFR in TNBC.
2) Change “from the patients” to “from patients”
3) Indicate the specific signatures for “Both epithelial and mesenchymal CSC”
Results
1) Change “unavailable dataset” to “unavailable datatsets” in 2.1
2) Change “did not inhibited” to “did not inhibit” in 2.2
Discussion
1) The induction of STAT3 is likely due to the induction of IL-6, these can be tied together.
2) Change “ex vivo patient tumors in experiments for clinical translation” to “ex vivo patient tumors in experiments or in in addition, these cell lines in vivo, for clinical translation”
Methods
1) Why not normalize to both 18S and GAPDH by geometric averaging?
2) Indicate the ANOVA post-test, and I assume this was a one-way ANOVA? Please clarify for completeness.
3) Why express the data as SE or SD? Should choose one and keep the same throughout, also indicating this in the Figure legends.
Figures, Tables, and Legends – None.
Reviewer 2 Report
The current manuscript by Suleiman et al examines the potential of using a combinatorial therapeutic approach for triple negative breast cancer (TNBC). TNBC is the most aggressive form of breast cancer with no available targeted therapies. The standard of care is the use of chemotherapeutics, such as cisplatin. However, cisplatin promotes EGFR- and hypoxia-mediated resistance of cancer stem cells, which have the potential to regrow and generate recurrent tumors. For this reason, the authors used gefinitib, an EGFR inhibitor, and metformin, an AMPK activator, in combination with cisplatin, to treat TNBC cell lines and ex vivo organotypic cultures of TNBC clinical samples and TNBC patient-derived xenograft tumors. The data overall support their hypothesis that cisplatin promotes survival of EGFR- and hypoxia-mediated cancer stem cells and that the concomitant use of gefinitib and metformin, together with cisplatin, significantly decreases survival of cancer stem cell cells, as well as of TNBC cells overall. The manuscript is well-written and puts forward the potential of a much needed combinatorial therapy that could be beneficial for TNBC patients. However, I do have a few comments that should be addressed, before the manuscript is considered for publication:
1) In addition to the two TNBC cell lines (MDA-MB-231 and SUM-149), the authors should use a normal cell line (e.g. MCF10A) as a control to treat with the same combination of therapeutics (as in Fig. 2), to demonstrate the specificity of this treatment for cancer cells and to exclude the possibility that this could be toxic to normal cells. This is really critical in order to propose that this kind of treatment can be used in the clinic to treat patients without serious adverse effects.
2) The authors use only n=5 ovarian cancer patients per case (cisplatin sensitive/resistant) in Suppl. Fig. 1. Were only 5 patients available or were these particular 5 selected? If there are more, using what criteria were these particular 5 selected?
3) It would strengthen the manuscript’s conclusions to show western blots of PDK1, ALDH1, (maybe also HIF1A, ALDH1) in Fig. 3 in all treatments, in addition to qPCRs, to demonstrate that indeed these markers were downregulated in the related treatments.
4) Minor comment: the Author contributions are missing
Round 2
Reviewer 2 Report
I am happy with the Authors' responses to my comments